# Untargeted Metabolomic Study for Urinary Characterization of Adult Patients with Phenylketonuria

**DOI:** 10.3390/ijms262411808

**Published:** 2025-12-06

**Authors:** Arnau Gonzalez-Rodriguez, Blanca Barrau-Martinez, Adriana Pané, Rosa Maria López Galera, Ester Tobias, Cristina Montserrat-Carbonell, Mariona Guitart-Mampel, Olga Jáuregui, Regina Roca-Vives, Judit Garcia-Villoria, Jose Cesar Milisenda, Ana Matas-Garcia, Maria de Talló Forga Visa, Pedro Juan Moreno Lozano, Gloria Garrabou, Mireia Urpi-Sarda, Rafael Llorach

**Affiliations:** 1Food and Lifestyle ExposOMics Research Group (FLExOMics-UB), Departament de Nutrició, Ciències de l’Alimentació i Gastronomia, Facultat de Farmàcia i Ciències de l’Alimentació, Campus de l’Alimentació de Torribera, Universitat de Barcelona (UB), 08921 Santa Coloma de Gramenet, Spain; agonzalezrodriguez@ub.edu (A.G.-R.); blancabarrau@ub.edu (B.B.-M.); 2Institut de Recerca en Nutrició i Seguretat Alimentària (INSA-UB), Campus de l’Alimentació de Torribera, Universitat de Barcelona (UB), 08921 Santa Coloma de Gramenet, Spain; 3Endocrinology and Nutrition Department, Hospital Clínic of Barcelona, 08036 Barcelona, Spain; pane@clinic.cat (A.P.); cmontse@clinic.cat (C.M.-C.); maforvi@gmail.com (M.d.T.F.V.); 4Adult Inborn Errors of Metabolism Unit, Hospital Clínic of Barcelona, 08036 Barcelona, Spain; rmlopez@clinic.cat (R.M.L.G.); etobiasb@ub.edu (E.T.); mguitart@recerca.clinic.cat (M.G.-M.); jugarcia@clinic.cat (J.G.-V.); jcmilise@clinic.cat (J.C.M.); anmatas@clinic.cat (A.M.-G.); pjmoreno@clinic.cat (P.J.M.L.); garrabou@clinic.cat (G.G.); 5Centro de Investigación Biomédica en Red de la Fisiopatología de la Obesidad y Nutrición (CIBEROBN), Instituto de Salud Carlos III, 28029 Madrid, Spain; 6Fundació Clínic per la Recerca Biomèdica (FCRB), Institut d’Investigacions Biomèdiques August Pi Sunyer (IDIBAPS), 08036 Barcelona, Spain; 7Division of Inborn Errors of Metabolism-IBC, Biochemistry and Molecular Genetics Department, Hospital Clínic of Barcelona, 08036 Barcelona, Spain; 8Centro de Investigación Biomédica en Red de Enfermedades Raras (CIBERER), Instituto de Salud Carlos III, 28029 Madrid, Spain; 9Inherited Metabolic Diseases and Muscle Disorders Research Laboratory, Cellex-IDIBAPS, Faculty of Medicine and Heath Sciences-University of Barcelona, 08036 Barcelona, Spain; 10Centres Científics i Tecnològics de la Universitat de Barcelona (CCiTUB), 08028 Barcelona, Spain; regiroca@ccit.ub.edu (R.R.-V.);; 11Centro de Investigación Biomédica en Red de Fragilidad y Envejecimiento Saludable (CIBERFES), Instituto de Salud Carlos III, 28029 Madrid, Spain

**Keywords:** phenylketonuria, non-targeted metabolomics, urine, metabolic profile, metabolite characterization, mass spectrometry

## Abstract

Phenylketonuria (PKU) is a rare inherited metabolic disorder caused by phenylalanine hydroxylase deficiency, leading to phenylalanine (Phe) accumulation and neurological dysfunction if untreated. While metabolomics holds promise for biomarker discovery in PKU, few studies have examined urinary metabolites using untargeted approaches. This study applied untargeted metabolomics using HPLC-QTOF-MS to analyze urine from 36 adult patients with PKU and 34 healthy controls. Biomarker Analysis was performed with MetaboAnalyst 6.0. A total of 73 significant metabolites (FDR < 0.05; VIP > 1) were identified, with 29 upregulated and 44 downregulated in PKU. A 23% of these metabolites were related to Phe metabolism, while 77% were associated with alterations across more than 10 metabolic pathways, including leucine and tryptophan metabolism, acylcarnitines, vitamins, and diet- or microbiota-derived compounds, among others. Specifically, upregulated metabolites with an AUC > 0.9 included several Phe-derived compounds, the nucleoside 8-hydroxy-7-methylguanine, and indole compounds (1H-indole-3-carboxaldehyde). Conversely, downregulated metabolites with an AUC > 0.9 included N-acetyl(iso)leucine and a heptenoylcarnitine isomer. The Random Forest-based model demonstrated enhanced predictive performance when integrating 10 metabolites, supporting their potential utility as biomarkers for PKU. These findings improve the biological understanding of metabolic disturbances beyond Phe, and may support the development of new therapeutic and dietary strategies.

## 1. Introduction

Phenylketonuria (PKU; OMIM #261600) is an inborn error of metabolism (IEM) caused by pathogenic variants in the gene encoding the enzyme phenylalanine hydroxylase (PAH) [1] with a global prevalence estimated at approximately 1:23,930 live births [2]. PAH is responsible for converting phenylalanine (Phe) to tyrosine (Tyr) using the cofactor tetrahydrobiopterin (BH4), molecular oxygen, and iron [3]. PAH deficiency leads to elevated Phe levels in the blood, urine, and brain [1], which, if untreated, result in variable intellectual disability, behavioral and psychiatric disturbances, motor dysfunction, and additional clinical features such as seizures and hypopigmentation of the skin [4]. Consequently, PKU is one of the most commonly included disorders in newborn screening programs worldwide. Many developmental impairments and associated symptoms become noticeable as the child grows [3].

The primary approach to managing PKU is the implementation of a low-Phe diet, initiated as early as possible to prevent irreversible neurological damage. Dietary management involves the restriction of natural protein sources to minimize Phe intake. To meet protein and nutrient requirements, patients are provided with synthetic protein substitutes (PS) composed of Phe-free amino acid mixtures. Additionally, glycomacropeptide (GMP)—a naturally low-Phe peptide derived from casein—may serve as an alternative protein source. These medical foods are typically fortified with essential vitamins, minerals, and trace elements to compensate for the nutritional deficiencies inherent in a low-protein diet [1,5]. Adherence to a strict diet is crucial for maintaining optimal metabolic control and preventing severe adverse effects in patients with PKU [4]. However, surveys indicate that dietary compliance is generally lower in adolescent and adult populations [6]. Lastly, some individuals with PKU may also benefit from adding pharmacological treatment with BH4, although the degree of responsiveness varies between individuals [1].

Metabolomics techniques are emerging as powerful tools for enhancing the efficiency of diagnosing IEMs, facilitating the global profiling of metabolites through mass spectrometry-based (MS) platforms [7]. Targeted metabolomics is employed at the first diagnostic level to identify patients with PKU, most often following abnormal findings in newborn screening programs. Diagnosis is confirmed by measuring elevated Phe concentrations and the Phe-to-Tyr ratio in plasma, alongside the analysis of amino acids in plasma and urine, and the assessment of organic acids and pterins in urine. In some cases, dihydropteridine reductase activity and BH4 metabolism studies are also performed to distinguish classical PKU from disorders of BH4 metabolism [4]. Beyond targeted assays, untargeted metabolomic strategies have the potential to discover novel biomarkers [8,9]. However, the lack of curated metabolite databases and spectral reference libraries may affect the precise identification of the detected compounds [10]. In the context of PKU, untargeted metabolomic analyses may have the potential to reveal distinct metabolic phenotypes, enabling the stratification of patients based on individualized metabolic signatures [11]. These insights could not only facilitate the identification of potential biomarkers but also contribute to the development of more precise tools for patient monitoring and personalized therapeutic strategies.

In this observational case–control study, we used an untargeted metabolomics approach to characterize the urinary metabolome of 36 adult patients with PKU and 34 healthy controls. The primary objective was to define the urinary metabolomic profile of patients with PKU and to identify potential biomarkers that reflect the biological complexity of the disease, providing novel insights through this untargeted analysis.

## 2. Results

### 2.1. Clinical Characteristics of the Study Population

This study included 70 individuals, comprising 36 adults diagnosed with classical PKU under nutritional counseling and 34 healthy controls. There were no statistically significant differences in demographics (age, sex) and anthropometric factors such as body mass index (BMI) between the two groups (controls and PKU group) (Table 1).

Dietary intake was comparable between groups in terms of total energy and overall protein consumption. As expected, natural protein intake was significantly lower in the PKU group due to dietary restrictions. In addition, differences in macronutrient composition were observed as fat intake was higher in the control group, whereas carbohydrate intake was significantly higher in the PKU group. These differences likely reflect distinct dietary habits influenced by the protein restriction inherent to PKU management, as previously reported in other studies comparing dietary habits of PKU and matched controls [12,13]. Moreover, intakes of vitamin B6 and vitamin E were higher among individuals with PKU, primarily due to supplementation from Phe-free amino acid formulas.

**Table 1 ijms-26-11808-t001:** Main characteristics of the study participants.

Analysis Cohort	PKU (*n* = 36)	Controls (*n* = 34)	*p*-Value ^3^
Anthropometric Data and Demographic Factors			
Age [mean (SD)], y	35.17 (9.77)	33.11 (9.44)	0.373
Female [*n* (%)]	15 (41.7)	21 (61.8)	0.093
BMI [mean (SD)], kg/m^2^	24.83 (4.27)	24.13 (3.35)	0.513
Classical PKU ^1^			-
Early dx, yes [*n* (%)]	28 (77.8)	-	
Late infant dx, yes [*n* (%)]	4 (11.1)	-	
Late adult dx, yes [*n* (%)]	4 (11.1)	-	
Adequate metabolic control ^2^	14 (39)	-	-
Dietary habits			
Energy intake [mean (SD)], kcal/day	1882.34 (501.03)	1859.02 (357.71)	0.827
PS energy intake [mean (SD)], kcal/day	453.27 (187.90)	-	-
Total protein intake [median (IQR)], g/day	78.97 (22.76)	78.16 (24.24)	0.224
Natural protein intake [mean (SD)], g/day	22.15 (12.12)	75.73 (18.90)	<0.001
Total protein intake [mean (SD)], %	17.93 (4.82)	16.61 (3.44)	0.202
Total fat intake [mean (SD)], %	28.81 (6.86)	44.57 (6.21)	<0.001
Total carbohydrate intake [median (IQR)], %	53.64 (11.03)	40.31 (9.18)	<0.001
Vitamin B6 intake [median (IQR)], mg/day	3.10 (1.41)	1.35 (1.34)	<0.001
Vitamin E intake [median (IQR)], mg/day	17.05 (8.35)	9.59 (4.89)	<0.001
Meat consumption, Yes [*n* (%)]	5 (13.9)	32 (94.1)	<0.001
L-carnitine supplementation	5 (13.9)	-	-
Biochemical parameters			
Phe [median (IQR)], µmol/L	847.60 (519.0)	55.95 (15.7)	<0.001
Tyr [median (IQR)], µmol/L	47.35 (26.9)	53.65 (28.1)	0.143
Total carnitine [median (IQR)], µmol/L	40.35 (11.7)	40 (12)	0.855
Free carnitine [mean (SD)], µmol/L	33.96 (8.72)	33.90 (7.54)	0.890
Leucine [median (IQR)], µmol/L	66.87 (12.88)	79.75 (23.65)	<0.001
Isoleucine [median (IQR)], µmol/L	28.70 (13.07)	31.05 (11.73)	0.062

^1^ Classical PKU is divided into ‘early diagnosed’ (<3 months of age), ‘late infant diagnosed’ (≥3 months–<7 years) [14] and ‘late adult diagnosed’ (>18 years). ^2^ Participants were classified with good metabolic control if their dried blood spot Phe levels were lower than 600 µmol/L [1,14]. ^3^ means two-tailed *p* value < 0.05. BMI, body mass index; dx: diagnosis; IQR, interquartile range; Phe, phenylalanine; PS, protein substitutes; SD, standard deviation; Tyr, tyrosine.

In terms of biochemical parameters, Phe levels were markedly elevated in adults with PKU compared to healthy controls (*p* < 0.001), whereas Tyr levels were slightly higher in controls, though this difference did not reach statistical significance. Levels of both free and total carnitine showed no significant variation between the two groups, with the presence of 5 subjects with PKU receiving additional L-carnitine supplementation (CARNICOR^®^ oral solution, 100 mg/mL (Alfasigma España S.L., Barcelona, Spain)).

### 2.2. Multivariate Analysis of Urine Samples

Untargeted metabolomics was used to investigate differences in metabolomic signatures between 36 treated adult patients with PKU and 34 matched healthy controls. A total of 4887 and 4984 features were obtained in both electrospray ionization (ESI) positive and ESI negative, respectively. A Principal Component Analysis (PCA) was performed to assess the overall metabolic variation among samples. The PCA results of urine samples revealed a distinct pattern between patients with PKU and healthy controls, particularly in the ESI+ model, suggesting metabolic differences between the two groups (Appendix A). An Orthogonal Partial Least Squares Discriminant Analysis (OPLS-DA) model was constructed (Figure 1) to explore the relationship between metabolite profiles and the two groups of participants (PKU vs. healthy controls). Both ESI+ and ESI− OPLS-DA models (Figure 1A and Figure 1B, respectively) displayed a clear separation between the two groups. A permutation test of 1000 numbers was carried out with MetaboAnalyst 6.0, and the results indicated that both models had good stability, reliability and quality [Q^2^ = 0.885, R^2^Y = 0.976, *p* < 0.001 for the ESI+ model (Appendix A) and Q^2^ = 0.865, R^2^Y = 0.977, *p* < 0.001 for the ESI− model (Appendix A)]. The Variable Importance in Projection (VIP) value (≥1) was combined with False Discovery Rate (FDR) values (<0.05) to determine if the metabolite feature was considered statistically significant. A total of 1078 features and 836 features were obtained for the ESI+ and ESI− models, respectively. From these features, 73 differential metabolites were identified.

### 2.3. Identification of Differential Metabolites

A detailed list of the identification parameters, including mass fragmentation, time retention and statistical values, is provided for each metabolite in Appendix A. In total, 29 metabolites were upregulated and 44 were downregulated in patients with PKU. Table 2 and Table 3 present these metabolites in decreasing order of VIP value for the upregulated and downregulated compounds, respectively. When categorized by metabolic pathway, 18 compounds were associated with dietary, microbiota, and micronutrient intake; 17 belonged to Phe metabolism; 14 to carnitine metabolism; 10 to tryptophan (Trp) metabolism; 3 to leucine (Leu) metabolism; and 2 to pteridine metabolism, among others.

#### 2.3.1. Phenylalanine and Phenylalanine Metabolism Metabolites

Phe (M01) and several compounds likely derived from its accumulation and metabolism were identified (M02–M17, M73). M01 showed the highest statistical significance (FDR 2.28 × 10^−18^) and the highest VIP value (3.75) (Appendix A). Tyr (M73), the main metabolite affected by PAH inactivity, was downregulated in urine samples from PKU patients, aligning with lower Tyr levels in the biochemical analysis (Table 1). Multiple Phe-derived compounds were found at elevated concentrations in individuals with PKU, including 2-hydroxyphenylacetic acid (M02) and phenyllactic acid (M09), as previously reported by Xiong et al. [15] γ-Glutamylphenylalanine (M05) and N-acetylphenylalanine (M11) were also elevated. N-lactoylphenylalanine (M08), a lactate-Phe conjugate, was also present in high concentrations. Carboxyethylphenylalanine isomers (M06 and M07), identified based on retention times and mass spectra, also showed higher peak intensities [16]. Two additional compounds, M12 and M13, were attributed to N-(ethoxyacetyl)phenylalanine isomers based on ChemSpider data [17]. Phe-hexose (M16) and phenylacetylglutamine (M17) were also found at increased levels. Several more Phe-related compounds (M03, M04, M10, M14, M15) were detected, although they have not previously been described in PKU.

#### 2.3.2. Nucleoside Compounds

8-Hydroxy-7-methylguanine (M18) is an endogenous methylated nucleoside derived from 7-methylguanine (7-MG) and is found in human biofluids. In this study, it has been found to be significantly upregulated in the PKU population compared to healthy controls.

#### 2.3.3. Pteridine Compounds

Pteridine compounds, specifically isoxanthopterin (M19) and dihydrobiopterin (M20), were identified at elevated levels in adults with PKU compared to healthy controls. Both metabolites have been commonly detected in previous PKU studies [18,19].

#### 2.3.4. Tryptophan and Tryptophan Metabolism Compounds

Several compounds (M21–M23, M46–M52) participating in the four Trp metabolic pathways (i.e., incorporation into proteins, indole metabolism, the kynurenine pathway, and the serotonin pathway) were altered in this cohort [20,21]. Subjects with PKU showed elevated levels of gut microbiota-derived indole metabolites, including 1H-indole-3-carboxaldehyde (M21), indolelactic acid (M22), and indoleacetic acid (M23), with M21 and M22 being consistent with previous findings [21,22]. In contrast, several compounds were downregulated, including free Trp (M47), N,N,N-trimethyltryptophan betaine (M46) which is a Trp betaine associated with legume intake [23,24], and C-Glycosyltryptophan (M49). Additionally, levels of 5-Hydroxyindoleacetic acid (M52), a serotonin pathway end-product, were reduced in subjects with PKU. Within the kynurenine pathway, kynurenine (M48) and kynurenic acid (M51) were also found to be decreased. Given the growing interest in these pathways in PKU, we further examined the relationship between urine and plasma levels for metabolites detected in both biological samples. Significant positive correlations were observed for 1H-indole-3-carboxaldehyde (r = 0.751, *p* < 0.001), indolelactic acid (r = 0.759, *p* < 0.001), N,N,N-trimethyltryptophan betaine (r = 0.834, *p* < 0.001), kynurenine (r = 0.414, *p* < 0.001), and Trp (r = 0.300, *p* = 0.0497), with consistent PKU–control differences in urine and plasma (*p* < 0.05). As expected from their shared metabolic origin, urinary kynurenine (M48) and urinary kynurenic acid (M51) were strongly correlated (r = 0.639, *p* < 0.001), and urinary kynurenic acid was also correlated with plasma kynurenine (r = 0.425, *p* < 0.001), supporting coordinated alterations in the kynurenine pathway across different matrices.

#### 2.3.5. Leucine-Derived Compounds

Three leucine-derived compounds (M30–M32) were identified at lower levels in the subjects with PKU, with N-acetyl(iso)leucine (M30) showing the highest FDR (5.51 × 10^−13^) and VIP value (3.40) among the downregulated metabolites (Appendix A). Consistent with these urinary findings, plasma leucine concentrations were significantly lower in patients with PKU compared with controls (*p* < 0.001), while plasma isoleucine only showed a trend (*p* = 0.062). In addition, we observed a positive correlation between plasma leucine levels and urinary N-acetyl(iso)leucine (M30) (r = 0.34, *p* = 0.009), supporting that this compound could be a leucine metabolite.

#### 2.3.6. Carnitine Metabolites

Although blood biochemical testing showed no significant differences in carnitine and total carnitine levels within the cohort, several urinary carnitine derivatives (M24, M33–M45) differed between individuals with PKU and healthy controls, while urinary free carnitine levels remained similar despite 5 patients receiving additional L-Carnitine supplementation (Table 1). Phenylacetylcarnitine (M24) was the only carnitine compound to be elevated in the PKU group, a finding first reported by Fischer et al. [25] in the urine of patients with PKU. This compound has also been observed to be upregulated in PKU in untargeted metabolomic studies [22]. Importantly, urinary phenylacetylcarnitine (M24) showed a positive correlation with its plasma levels (r = 0.588, *p* < 0.001), also being upregulated in patients with PKU. The other acylcarnitines identified in this study showed lower levels in patients with PKU. In particular, the acylcarnitine M36—putatively annotated as undecanoylcarnitine, dimethylnonanoylcarnitine, or methyldecanoylcarnitine—was reduced in both urine and plasma of patients with PKU, a pattern further supported by a positive correlation between the two matrices (r = 0.361, *p* = 0.002). This indicates that the decrease observed in urine is mirrored by the same direction of change in plasma.

#### 2.3.7. Micronutrients and Dietary Metabolites

Several micronutrient-related metabolites were elevated in the PKU group, including two vitamin E metabolites (M26 and M27), the vitamin B6 metabolite 4-Pyridoxic acid (M25) and the vitamin B5 metabolite pantothenic acid (M28). These findings align with our reported higher intake levels of vitamins B6 and E among individuals with PKU (Table 1).

Compounds of dietary intake also revealed group differences. Metabolites associated with animal protein consumption, i.e., 1-Methylhistidine (M53), 3-Methylhistidine (M55), 5-Aminovaleric acid betaine (M54) and creatine (M56), were present in lower levels in the PKU group. Additionally, the advanced Maillard reaction product pyrraline (M65) was reduced. Several gut microbiota-derived phenolic compounds, including urolithin B glucuronide (M58), urolithin A glucuronide (M61), enterolactone glucuronide (M60), and dihydroxy-H-indole glucuronide isomers (M59, M62), were also present at lower levels. Lastly, caffeine-related metabolites (M67–M70), derived from coffee, tea, and cocoa, were downregulated in individuals with PKU.

#### 2.3.8. Glycine Metabolites and Other Amino Acid Compounds

Two glycine compounds (M63 and M64) were found to be downregulated in the PKU group compared to controls. Additionally, several other amino acid-related compounds (M66, M71, M72) were decreased in adults with PKU.

### 2.4. Enrichment Analysis of Differential Metabolites

To explore the potential pathways and functions associated with the differential metabolites identified between patients with PKU and healthy individuals, we performed a quantitative enrichment analysis with MetaboAnalyst 6.0, selecting the Kyoto Encyclopedia of Genes and Genomes (KEGG) pathway (Figure 2) and urinary metabolite sets for disease signature analysis (Appendix A). Of the 73 metabolites included, 36 metabolites were recognized by MetaboAnalyst 6.0 although not all of them had a KEGG ID (27 out of the 36).

Enrichment analysis from KEGG pathway analysis (Figure 2) illustrated that amino acids are altered between patients and controls, especially aromatic amino acids (Phe, Tyr, and Trp). Phe metabolism and Phe, Tyr and Trp biosynthesis were the most significant pathways (1.32 × 10^−24^ and 1.84 × 10^−24^ FDR, respectively) and had the highest enrichment ratio (45 and 36, respectively). In Trp metabolism, Trp (M47) and its kynurenine pathway metabolites (M48 and M51) as well as serotonin pathway metabolite (M52) were decreased in the PKU group, while microbial metabolites of tryptophan (M21–M23) were markedly elevated in adult patients with PKU. Additional pathways were also altered due to the metabolites identified.

### 2.5. ROC Analysis

The Biomarker Analysis module in MetaboAnalyst has been used to evaluate the 73 identified metabolites (Table 2 and Table 3) as potential biomarkers. A total of 15 metabolites exhibited Area Under the Curve (AUC) values greater than 0.9 after classical univariate Receiver Operating Characteristic (ROC) curve analysis (Figure 3), indicating a strong diagnostic potential. Among these, 8-hydroxy-7-methylguanine (M18) showed the highest AUC value, followed by Phe (M01) and other Phe-related metabolites. Additional compounds such as N-acetyl(iso)leucine (M30), isoxanthopterin (M19), 1H-indole-3-carboxaldehyde (M21), and heptenoylcarnitine (M33) also demonstrated high AUC values. Notably, M30 and M33 were downregulated in the PKU group. Sensitivity and specificity values for each metabolite are available in Appendix A.

The Models built using Random Forest (RF) (Figure 4A) improved predictive performance when combining 3, 5 and 10 metabolites. This approach underscores the value of using a metabolite panel rather than a single compound [26], as it captures a more comprehensive view of PKU’s metabolic impact. Figure 4D displays the selection frequency for the top-3, top-5, and top-10 metabolites from the respective models (Figure 4B,D).

These results illustrate the metabolic and biological complexity of PKU, with metabolites belonging to diverse pathways (Trp metabolism, Leu metabolism, purine metabolism, and Phe metabolism), all contributing significantly to the urinary metabolomic signature of PKU.

## 3. Discussion

This study offers a detailed urinary metabolomic comparison between adults with PKU and healthy controls, revealing a wide range of metabolic alterations. Although Phe and Phe-related compounds accounted for 23% of the identified metabolites, the majority of identified metabolites (77%) belonged to other pathways, including those related to several amino acids, acylcarnitines, micronutrients, and dietary and microbial metabolites. These findings highlight the complex metabolic impact of PKU and provide biological insights that may influence disease progression and clinical outcomes.

The diversity of Phe-related metabolites identified in this study reflects the broad biochemical consequences of phenylalanine accumulation in PKU. While some compounds like 2-hydroxyphenylacetic acid (M02), phenyllactic acid (M09), N-acetylphenylalanine (M11), and phenylacetylglutamine (M17) have already been proposed as markers in prior studies [15], others, such as N-lactoylphenylalanine (M08) and the Amadori rearrangement product Phe-hexose (M16) [27] may suggest additional metabolic disruptions with potential relevance. γ-Glutamylphenylalanine (M05) and N-acetylphenylalanine (M11) repeatedly have been reported as upregulated in blood samples from patients with PKU [8,22,28] with M05 being found in urine samples by Peck and Pollitt in 1979 [29]. Phenylacetylglutamine (M17) has also been proposed as a non-invasive urinary biomarker, as Andrade et al. [30] found a significant correlation between urine M17 levels and circulating Phe levels in patients with PKU and hyperphenylalaninemia (HPA). N-lactoylphenylalanine (M08), a conjugate of lactate and Phe, was identified in high concentrations for the first time in the PKU population by Jansen et al. [31] in 2015. This compound has recently been suggested by van Wegberg et al. [28] as a marker of clinical interest, as it shows a stronger association with working memory and mental health outcomes than Phe levels alone. The detection of novel or rarely described Phe derivatives (e.g., M03, M04, M10, M14, M15) further expands the known urinary signature of PKU.

Nevertheless, the untargeted approach also revealed significant alterations in several metabolites (M18, M21, M30 and M33). These findings confirm that, although Phe metabolism remains central to PKU pathophysiology, a large proportion of the urinary metabolomic alterations (77% of the identified metabolites) originate from other pathways, reinforcing that PKU involves a broader metabolic complexity beyond Phe, as previously proposed by other authors [28].

8-Hydroxy-7-methylguanine (M18) was identified as one of the most discriminant metabolites in our cohort, showing the highest AUC value (0.99) in the univariate ROC analysis (Figure 3A) and being consistently selected in the Random Forest models (Figure 4). This compound has been previously detected in human urine by Liu et al. [32] in their study among healthy adults and children. Biochemically, 8-hydroxy-7-methylguanine originates from the oxidation of 7-MG, a reaction catalyzed by the enzyme xanthine oxidase [33]. Since 7-MG has been described as a marker of DNA damage, as it can induce DNA strand breaks [34], its oxidation may reflect altered nucleotide metabolism or oxidative stress. In this sense, Dobrowolski et al. [35] observed aberrant DNA methylation in patients with PKU, with a more extensive methylation in patients with high Phe exposure [35]. To our knowledge, this is the first report of elevated urinary 8-hydroxy-7-methylguanine (M18) in PKU, suggesting that purine metabolism and oxidative processes may contribute to the metabolic complexity of PKU and indicates the need for further research to assess its potential as a biomarker.

An important biological insight of our cohort is the alteration of Trp metabolism, with both up- and downregulated metabolites (14% of identified metabolites) across its main pathways. Trp catabolism can be divided in: the indole and tryptamine pathway, serotonin pathway, and kynurenine pathway [21]. Indole metabolites (M21–M23) concentrations, derived from microbiota, were higher in patients with PKU, with M21 and M22 being consistent with previous findings [21,22]. Notably, 1H-indole-3-carboxaldehyde (M21) showed high potential as a marker in both univariate (Figure 3) and multivariate ROC curves analyses (Figure 4D). This reinforces growing evidence in gut–brain axis interactions in PKU, where microbiota-derived metabolites might play a role in clinical manifestations [21]. Reduced levels of 5-Hydroxyindoleacetic acid (M52), a urinary serotonin breakdown product, in patients with PKU suggest alterations in the serotonin pathway [1], a minor component of Trp metabolism [36]. Lower levels of M52 have been associated with depression, behavioral disturbances [37], and increased migraine occurrence [38], which may be neurocognitive factors potentially relevant to PKU. The kynurenine pathway, which accounts for the majority of the catabolism of ingested Trp not used for protein synthesis [36,39], showed distinct alterations, with kynurenine (M48) and kynurenic acid (M51) being reduced in the PKU group. This pathway has been associated with inflammation and psychiatric disorders, as kynurenines have neuromodulatory properties [36]. Kynurenine (M48) is preferentially metabolized into various compounds, including the neurotoxin quinolinic acid which was not identified as a statistically significant feature in our analyses. Alternatively, kynurenine (M48) can be converted into kynurenic acid (M51), known for its neuroprotective effects [40]. Schoen et al. [22] also reported plasma reduced levels of kynurenine (M48) and Trp (M47) in females with PKU and poor metabolic control. Boulet et al. [40] similarly observed a disruption in plasma Trp metabolism in patients with PKU, with downregulation of kynurenine (M48), upregulation of kynurenic acid (M51), and no significant differences in Trp (M47). In our metabolomic analysis, we observed reduction in Trp itself and in related compounds such as N,N,N-trimethyltryptophan betaine (M46) and C-Glycosyltryptophan (M49), which may be linked to the dietary restrictions of PKU. These results highlight a complex disruption of Trp metabolism in PKU, likely influenced by both dietary and microbial factors.

Among the metabolites identified as potential biomarkers beyond Phe metabolism, N-acetyl(iso)leucine (M30) was found at lower levels in patients with PKU, together with other (iso)leucine derivates (M31, M32). Leu and isoleucine (Ile) are essential large neutral amino acids (LNAAs), primarily found in animal protein sources, that compete with Phe for brain uptake [41]. Both Leu and Ile have been previously found in metabolomic studies, analyzing plasma samples, to be downregulated in PKU [22,42]. Elevated blood LNAA concentrations may reduce Phe transport into the brain [41], and newer dietary options for PKU, such as GMP, contain higher LNAA levels compared to other PS [6]. Since Leu and Ile are essential for protein synthesis and energy regulation, and protein sources in PKU are largely limited to amino acid mixtures, patients with PKU may be at risk for muscle mass deficits [43]. Additional research is needed to clarify how alterations in urinary leucine-derived metabolites relate to systemic LNAA metabolism and nutritional status.

Acylcarnitines represented another major group of altered metabolites in this PKU population (19% of total metabolites). We observed increased levels of both urinary and plasma phenylacetylcarnitine (M24) in patients with PKU, as well as decreased levels of 13 urinary acylcarnitines (M33–M45), with decreased plasma levels of M36. Although free carnitine deficiency is commonly reported in PKU and has been associated with increased oxidative stress [44,45,46], no such deficiency was observed in our cohort, which may have been influenced by supplementation or cohort characteristics. Previous studies describing acylcarnitine alterations in PKU have primarily analyzed plasma [22] or dried blood spots [47,48], reporting both upregulation (including phenylacetylcarnitine) [22] and downregulation of various acylcarnitines species [47,48]. Changes in acylcarnitines profiles have been linked to impaired energy metabolism [48] and have been proposed to play a role in neurotransmission [49]. Further investigation into the role of specific carnitine derivatives identified is warranted to clarify their contribution to PKU pathophysiology.

Dietary-related metabolites accounted for approximately 25% of the identified compounds, reflecting both the dietary restrictions and the use of PS. Four vitamin-related metabolites (M25–M28) were upregulated in patients with PKU, mainly due to the use of fortified Phe-free amino acid formulas [1]. A recent systematic review and meta-analysis by Bokayeva et al. [50] examined studies on early-treated patients with PKU managed exclusively with dietary treatment, comparing their vitamin levels in blood or plasma to those of healthy control groups. The analysis found no significant differences in the levels of vitamin A, E, B6, B12 or 25-hydroxyvitamin D, although folate and 1,25-dihydroxyvitamin D concentrations were higher in the PKU group. Pantothenic acid (M28) was not assessed in the systematic review due to the lack of studies. Fourteen dietary compounds (M53–M56, M58–M62, M65, M67–M70) were downregulated. The reduced excretion of dietary biomarkers related to animal protein (methylhistidines, 5-aminovaleric acid betaine, and creatine; M53–M56) [51,52,53,54] reflects the restricted natural protein intake typical in PKU dietary management. This pattern aligns with the dietary data, showing lower meat consumption and reduced natural protein intake among individuals with PKU (Table 1), despite similar total protein and energy intake between groups. It is important to note that the diet of the control population was consistent with the macronutrient distribution reported for the general Spanish adult population [55,56]. Creatine (M56) also plays a key role in energy metabolism [57], which has been found to play a role in PKU pathophysiology [46], and low urinary levels compared to healthy controls have also been previously reported [58]. The lower levels of microbiota-derived phenolic compounds (M58–M62) [51,59,60] could indicate altered gut microbiome activity in PKU, consistent with studies describing compositional and functional differences in the gut microbiota of this population [61,62]. Additionally, the downregulation of caffeine metabolites (M67–M70) may reflect dietary choices or caution around aspartame-containing products, which are typically avoided in PKU [6]. These results emphasize the complex interaction between dietary restrictions, supplementation and metabolism in PKU.

To our knowledge, few studies have performed ROC analyses in search of new potential biomarkers in blood [63] or in urine [15,64]. Bao et al. [64] evaluated urinary pteridines to distinguish individuals with HPA from those with BH4 synthetase deficiency, and reported isoxanthopterin levels (M19) as a discriminant metabolite, which in our study achieved an AUC of 0.94 in univariate ROC analyses. Recently, Şahiner et al. [63] also performed ROC analyses with blood samples in children with PKU and observed several metabolites (Phe and phe-related metabolites, and other amino acids) to have a strong discriminatory power. Xiong et al. [15], proposed several Phe-derived metabolites as diagnostic biomarkers using a gas chromatography coupled to mass spectrometry-based metabolomic approach with urine samples. According to their ROC analyses, the compounds included Phe, 2-hydroxyphenylacetic acid, N-acetylphenylalanine and phenylacetylglutamine, which showed AUC values close to 0.6, while phenyllactic acid reached an AUC of 0.97. In our analyses, Phe (M01), 2-hydroxyphenylacetic acid (M02), and N-acetylphenylalanine (M11) exhibited AUCs above 0.9, whereas phenyllactic acid (M09) and phenylacetylglutamine (M17) showed AUCs of 0.88 and 0.71, respectively.

## 4. Materials and Methods

### 4.1. Subjects and Study Design

Participants were recruited from the Adult Inherited Metabolic Disorders Unit at the Hospital Clínic of Barcelona (Barcelona, Catalonia, Spain) between 2021 and 2023. A total of 70 individuals (36 patients with PKU and 34 healthy controls) were included in this study. Patients with PKU were under nutritional counseling and clinical monitoring, following dietary recommendations that included the use of PS and a low-Phe diet [1]. None of the patients were receiving BH4 treatment. To ensure comparability, healthy participants in the control group were matched to individuals with PKU according to age, sex, and BMI.

Study design and procedures were approved by the Bioethics Committee of the University of Barcelona (IRB00003099) (approved on 5 October 2020) and Hospital Clínic of Barcelona (HCB/2020/0552) (approved on 25 June 2020) and was conducted in accordance with the Declaration of Helsinki. This study is listed on the ISRCTN registry (www.isrctn.com) under the ID number ISRCTN12620764, https://www.isrctn.com/ISRCTN12620764 (accessed on 16 July 2025). Participants were included if they were >18 years old and genetically diagnosed with classic PKU or HPA. Subjects were excluded if they met any of the following criteria: an intelligence quotient below 70, as assessed by the Wechsler Adult Intelligence Scale—Fourth Edition; current pregnancy or intent to become pregnant during the study period; active cancer; severe chronic liver disease; a recent history (within the previous six months) of an acute cardiovascular event; or elevated serum creatinine levels (>2.0 mg/dL). All participants in this study provided their written informed consent.

Morning urine and plasma samples were collected after an overnight fasting period to ensure standardized metabolic conditions across participants. All samples were immediately stored at −80 °C until analysis.

Dietary habits were assessed through 4-day food records, analyzed using the Spanish nutritional database Odimet^®^ (https://www.odimet.es/; accessed on 1 September 2024) to estimate the daily intake of macro- and micronutrients.

### 4.2. Sample Preparation

Urine samples were thawed overnight at 4 °C before analysis. Samples were vortex mixed and centrifuged for 5 min at 12,000× *g* and 4 °C. A 150 µL aliquot of the supernatant was diluted with 150 µL of Milli-Q water and the samples were arranged in a 96-well plate matrix format (8 × 12 grid structure) for high-performance liquid chromatography coupled to an Ion Mobility quadrupole time-of-flight mass spectrometer (HPLC-QTOF-MS) analysis. QC samples [65] were used: QC1 were Milli-Q water samples, QC2 were a standard mixture solution of different compounds at 5 ppm in H_2_O, QC3 consisted of repeated injections of a urine sample from a healthy individual following a standard diet and QC4 were reinjections of samples between and across plates. Finally, the injection of samples was randomized to avoid possible bias. Plasma samples were additionally analyzed to complement the interpretation of urinary findings. Details of plasma sample preparation are provided in the Appendix A [66,67].

### 4.3. HPLC-QTOF-MS Analysis

Chromatography was performed on an Agilent 1290 II Infinity system (Agilent technologies, Santa Clara, CA, USA) using an RP 18 Luna 5 µm, 50 × 2.0 mm column (Phenomenex, Torrance, CA, USA). The mobile phase consisted of (A) H_2_O 0.1% HCOOH and (B) ACN 0.1% HCOOH. The flow rate was 600 µL/min, and the injection volume was 5 µL for both urine samples and QCs. A linear gradient with the following proportions (*v*/*v*) of phase B (t, %B) was used: (0, 1), (4, 20), (6, 95), (7.5, 95), (8, 1), (12, 1) [65]. The re-equilibration time of the column was 4 min. The HPLC system was coupled with an Agilent 6560 Ion Mobility Q-TOF LC/MS equipped with a Dual Agilent Jet Stream Technology ESI source (Agilent technologies, Santa Clara, CA, USA). The MS acquisition was performed in positive and negative modes and TOF Scan mode (50–1200 m/z). Operation parameters for positive and negative modes were as follows: gas temperature, 300 (°C); drying gas flow, 5 (L/min); nebulizer gas pressure, 35 (psi); sheath gas temperature, 350 (°C); sheath gas flow, 11 (L/min); Capillary voltage, 3500 (V); nozzle voltage, 1000 (V); fragmentor voltage, 400 (V). Additional information about the parameters can be found in Appendix A.

### 4.4. Data Processing and Statistical Analysis

Statistical analysis for the clinical characteristics of the participants and plasma metabolites was conducted using the SPSS software package (SPSS 27.0, IBM, Armonk, NY, USA). A two-tailed *p*-value of <0.05 was considered statistically significant. The Shapiro–Wilk test was used to assess normality, and non-normally distributed variables were log 10 transformed prior to analysis. Differences in clinical characteristics and plasma metabolites between individuals with PKU and healthy controls were assessed using Student’s *t*-test for normally distributed continuous variables, the Mann–Whitney U test for non-normally distributed variables, and the chi-square test for categorical variables. Spearman correlation analyses were performed between urinary levels and their corresponding plasma levels to evaluate its relationship and assess whether urinary alterations reflected plasma metabolic changes.

After HPLC-QTOF-MS analysis, raw chromatographic and spectral data were first analyzed using MassHunter Qualitative Analysis 10.0 and then processed with MassHunter Profinder B.10.0.02. Data was processed adapting methods from Navarro et al. [68] Profinder software 10.0.02 used the Recursive Feature Extraction algorithm to generate a list of unique features by time and mass alignment and additional filtering criteria [69]. Appendix A describes the Profinder parameters to perform the feature extraction. Mass Profiler Professional 15.1 was used to transform raw data in txt files, adapting Navarro et al. [68] procedures (Appendix A). The table generated was cube root-transformed and range-scaled in MetaboAnalyst 6.0 software [70]. This dataset was used for samples and QC quality assessment, statistical analysis and metabolomics visualization. A Student’s *t*-test and an OPLS-DA, after a 1000-response permutation test, were applied to identify statistically significant features that differentiated patients with PKU from healthy controls. Features were selected based on FDR threshold of less than 0.05 and a VIP score greater than 1.

### 4.5. Metabolite Identification

Metabolite features with VIP values higher than 1 and FDR less than 0.05 in positive and negative ionization modes were selected. Features were initially tentatively annotated based on a mass accuracy threshold of ±5 mDa from their theoretical mass. This process used PermutMatrix [71], MetFrag (https://msbi.ipb-halle.de/MetFrag/; accessed on 1 June 2025), Human Metabolome DataBase (HMDB) [72], and Metabolite Automatic Identification Toolkit (MAIT) R’s package [73]. Significant features were identified using MS/MS experiments which were performed with a collision energy of 15 or 30 V as needed. MS/MS spectra were used for compound identification, primarily through METLIN database matching with MassHunter PCDL Manager B.08.00, and further supported by prior literature and in silico fragmentation via Metfrag. Reference standards were used for compound confirmation (first level of metabolite identification) when available [74]. Finally, Metfrag, ChemDraw 22.2.0 and MassHunter Qualitative Analysis were used to generate compound and fragment formulas for molecular structural elucidation.

### 4.6. Enrichment and Biomarker Analysis

The Enrichment Analysis and Biomarker Analysis modules in MetaboAnalyst 6.0 were used to identify affected metabolic pathways and to perform ROC curves analysis, respectively. Both analyses were performed using the most statistically significant features. The previous transformed and scaled dataset was used for the quantitative enrichment analysis with KEGG [75] pathway sets to determine the metabolic pathways affected.

Classical univariate and multivariate ROC curve analyses were conducted to identify potential biomarkers. Model performance was evaluated by the AUC and by sensitivity and specificity, calculated at the optimal cut-off point. The optimal cut-off was determined as the minimum distance to the top-left corner of the ROC plot [76]. In the multivariate ROC analysis, a RF algorithm was applied for biomarker selection. To validate the models, MetaboAnalyst 6.0 employs Monte Carlo cross-validation with balanced sub-sampling, repeating the procedure multiple times to calculate model performance and confidence intervals [70,77].

## 5. Conclusions

This study performed an untargeted metabolomics approach to analyze the urinary metabolomic differences between 36 adult patients with PKU and 34 healthy controls. A total of 73 differential metabolites were identified, with 29 upregulated and 44 downregulated in the PKU group. While 23% of the metabolites were related to Phe metabolism, 77% of the identified metabolites belonged to other altered pathways, including Trp, Leu, carnitine, purine, pteridine and vitamin metabolism, as well as dietary and microbial metabolites.

ROC analyses revealed several metabolites with strong discriminative power (AUC > 0.9). Upregulated metabolites in the PKU group included Phe and Phe-derived compounds, such as γ-Glutamylphenylalanine, 2-hydroxyphenylacetic acid, and carboxyethylphenylalanine isomers, as well as nucleoside (8-hydroxy-7-methylguanine), pteridine (isoxanthopterin), and indole (1H-indole-3-carboxaldehyde) compounds. Among the downregulated metabolites, N-acetyl(iso)leucine showed a high discriminative capacity both individually and in the multivariate RF models, while heptenoylcarnitine exhibited better performance in the univariate analysis.

These results reinforce the potential of urinary metabolomic characterization to enhance the biological understanding of PKU and its metabolic complexity, and may support the development of new therapeutic and dietary strategies. These findings highlight promising avenues for intervention; therefore, further metabolomic research is essential to validate these potential biomarkers and to better understand their biological and clinical relevance in PKU, with the ultimate goal of enhancing patient outcomes and quality of life.

## Figures and Tables

**Figure 1 ijms-26-11808-f001:**
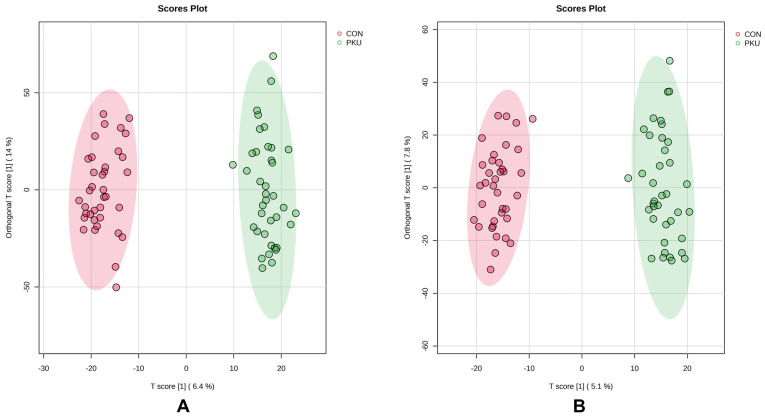
OPLS-DA scores plot of population samples in ESI+ (**A**) and ESI− (**B**). Red and green dots are the samples from healthy controls and individuals with PKU, respectively.

**Figure 2 ijms-26-11808-f002:**
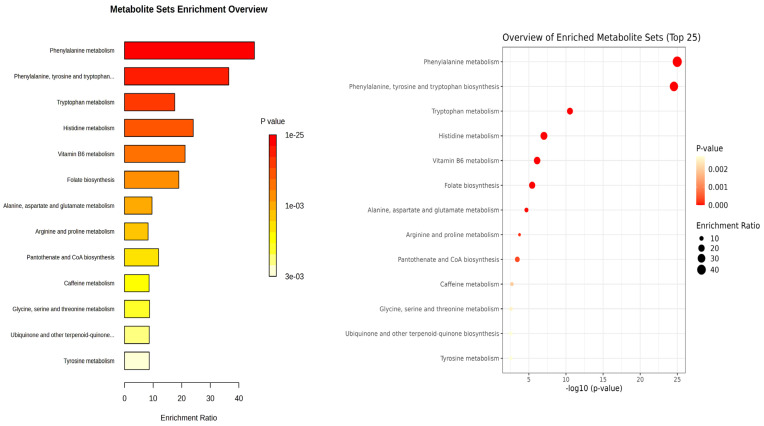
Quantitative Enrichment Analysis of differential metabolites using KEGG database. Enrichment ratio is determined as the number of hits within a specific metabolic pathway divided by the expected number of hits.

**Figure 3 ijms-26-11808-f003:**
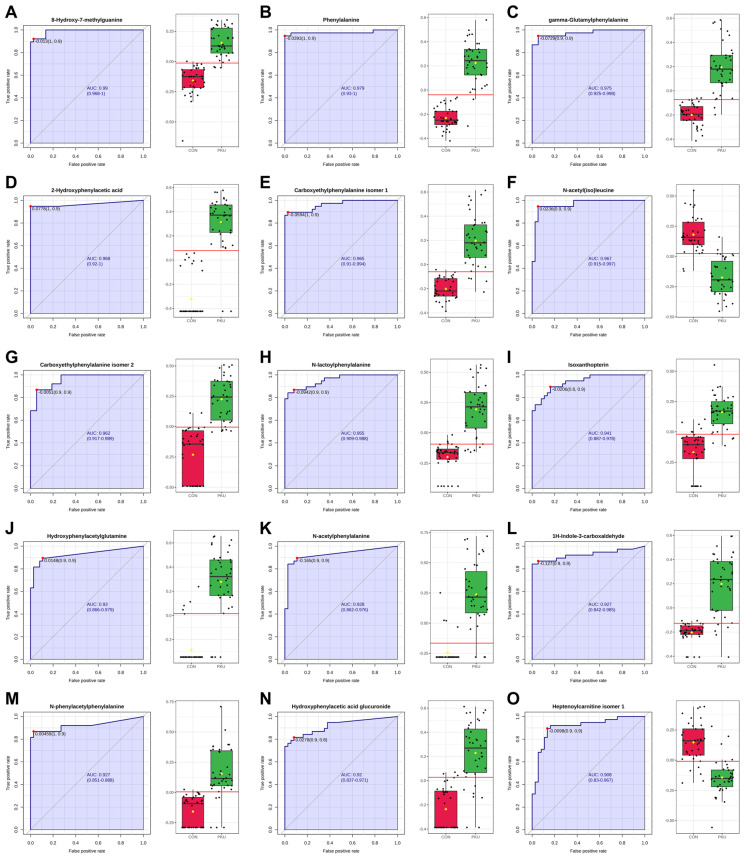
ROC analysis of potential biomarkers and normalized abundance in control (red) vs. PKU (green). Red dot and red line correspond to the optimal cutoff value of the ROC curve. Metabolites are sorted from the highest to the lowest AUC value. (**A**) M18, 8-hydroxy-7-methylguanine. (**B**) M01, phenylalanine. (**C**) M05, γ-Glutamylphenylalanine. (**D**) M02, 2-hydroxyphenylacetic acid. (**E**) M06, carboxyethylphenylalanine isomer 1. (**F**) M30, N-acetyl(iso)leucine. (**G**) M07, carboxyethylphenylalanine isomer 2. (**H**) M08, N-lactoylphenylalanine. (**I**) M19, isoxanthopterin. (**J**) M03, hydroxyphenylacetylglutamine. (**K**) M11, N-acetylphenylalanine. (**L**) M21, 1H-indole-3-carboxaldehyde. (**M**) M10, N-phenylacetylphenylalanine. (**N**) M04, hydroxyphenylacetic acid glucuronide. (**O**) M33, heptenoylcarnitine isomer 1.

**Figure 4 ijms-26-11808-f004:**
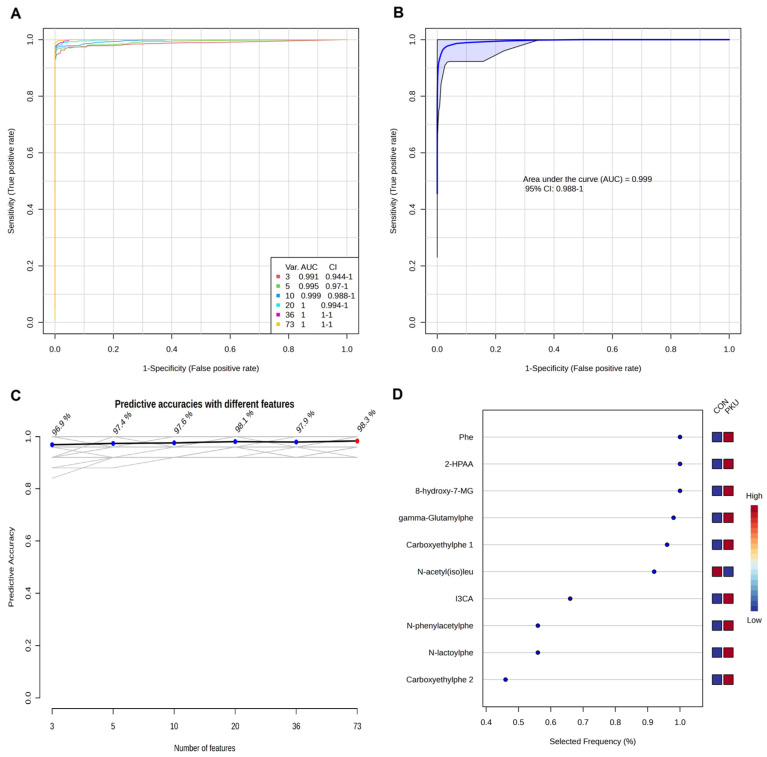
Biomarker prediction by Multivariate ROC curve based exploratory analysis by Random Forest algorithm. (**A**) ROC curves for all the models. (**B**) ROC curve for the model with 10 potential biomarkers. (**C**) Predictive accuracies with different number of features. (**D**) Top-10 most selected features for the model with 10 features. Selected Frequency of 1 means being selected 100% of the time by cross-validation. Phe, phenylalanine (M01); 2-HPAA, 2-hydroxyphenylacetic acid (M02); 8-Hydroxy-7MG, 8-hydroxy-7-methylguanine (M18); gamma-Glutamylphe, gamma-Glutamylphenylalanine (M05); Carboxyethylphe 1, carboxyethylphenylalanine isomer 1 (M06); N-acetyl(iso)leu, N-acetyl(iso)leucine (M30); I3CA, 1H-indole-3-carboxaldehyde (M21); N-phenylacetylphe, N-phenylacetylphenylalanine (M10); N-lactoylphe, N-lactoylphenylalanine (M08); Carboxyethylphe 2, carboxyethylphenylalanine isomer 2 (M07).

**Table 2 ijms-26-11808-t002:** List of upregulated urinary metabolites in adults with PKU ranked by VIP values.

M	Metabolite	VIP	Biological Source/Pathway
M01	Phenylalanine	3.75	Phenylalanine metabolism
M02	2-Hydroxyphenylacetic acid	3.69	Phenylalanine metabolism
M03	Hydroxyphenylacetylglutamine	3.47	Phenylalanine metabolism
M18	8-Hydroxy-7-methylguanine	3.25	Purine metabolism
M04	Hydroxyphenylacetic acid glucuronide	3.20	Phenylalanine metabolism
M05	γ-Glutamylphenylalanine	3.14	Phenylalanine metabolism
M06	Carboxyethylphenylalanine isomer 1	3.09	Phenylalanine metabolism
M07	Carboxyethylphenylalanine isomer 2	3.07	Phenylalanine metabolism
M08	N-lactoylphenylalanine	3.02	Phenylalanine metabolism
M09	Phenyllactic acid	3.01	Phenylalanine metabolism
M10	N-phenylacetylphenylalanine	3.00	Phenylalanine metabolism
M11	N-acetylphenylalanine	2.97	Phenylalanine metabolism
M19	Isoxanthopterin	2.97	Pteridine pathway
M21	1H-Indole-3-carboxaldehyde	2.96	Tryptophan metabolism
M12	N-(ethoxyacetyl)phenylalanine isomer 1	2.93	Phenylalanine metabolism
M22	Indolelactic acid	2.75	Tryptophan metabolism
M13	N-(ethoxyacetyl)phenylalanine isomer 2	2.73	Phenylalanine metabolism
M23	Indoleacetic acid	2.42	Tryptophan metabolism
M14	Hydroxyphenylacetic acid sulfate	2.36	Phenylalanine metabolism
M24	Phenylacetylcarnitine	2.30	Acylcarnitine pathway
M25	4-Pyridoxic acid	2.10	Vitamin B6 metabolism
M20	Dihydrobiopterin	2.04	Pteridine pathway
M26	α-CEHC glucuronide	2.04	Vitamin E metabolism
M15	N-phenylacetylglutamic acid	2.02	Phenylalanine metabolism
M27	α-CEHC	1.93	Vitamin E metabolism
M28	Pantothenic acid	1.71	Pantothenate biosynthesis
M16	Phe-hexose	1.64	Phenylalanine metabolism
M29	1-Pyrroline-5-carboxylic acid	1.39	Glutamate metabolism
M17	Phenylacetylglutamine	1.35	Phenylalanine metabolism

**Table 3 ijms-26-11808-t003:** List of downregulated urinary metabolites in adults with PKU ranked by VIP values.

M	Metabolite	VIP	Biological Source/Pathway
M30	N-acetyl(iso)leucine	3.40	Leucine, isoleucine and valine metabolism
M33	Heptenoylcarnitine isomer 1	2.77	Acylcarnitine pathway
M46	N,N,N-trimethyltryptophan betaine	2.71	Tryptophan metabolism
M53	1-Methylhistidine	2.64	Animal protein consumption
M57	Dihydroxybenzoic acid isomer	2.60	Drug-topical agent
M34	Hydroxyundecanoylcarnitine isomer	2.57	Acylcarnitine pathway
M58	Urolithin B glucuronide	2.48	Phenolic compound
M35	Octanoylcarnitine or methylheptanoylcarnitine or valproylcarnitine	2.47	Acylcarnitine pathway
M36	Undecanoylcarnitine or dimethylnonanoylcarnitine or methyldecanoylcarnitine	2.36	Acylcarnitine pathway
M37	Heptenoylcarnitine isomer 2	2.36	Acylcarnitine pathway
M54	5-Aminovaleric acid betaine	2.35	Animal protein consumption
M63	Hepteneoylglycine isomer	2.33	Glycine compound
M55	3-Methylhistidine	2.30	Animal protein consumption
M38	Octanoylcarnitine or methylheptanoylcarnitine or valproylcarnitine	2.24	Acylcarnitine pathway
M39	Heptanoylcarnitine or methylhexanoylcarnitine	2.20	Acylcarnitine pathway
M40	Decanoylcarnitine or methylnonanoylcarnitine	2.17	Acylcarnitine pathway
M59	Dihydroxy-H-indole glucuronide isomer 1	2.09	Phenolic compound
M65	Pyrraline	2.09	Food component
M60	Enterolactone glucuronide	2.03	Phenolic compound
M47	Tryptophan	1.99	Tryptophan metabolism
M48	Kynurenine	1.98	Tryptophan metabolism
M61	Urolithin A glucuronide	1.96	Phenolic compound
M62	Dihydroxy-H-indole glucuronide isomer 2	1.96	Phenolic compound
M41	Dodecenoylcarnitine	1.93	Acylcarnitine pathway
M42	Oxononanoylcarnitine or hydroxynonenoylcarnitine isomers	1.92	Acylcarnitine pathway
M43	Oxononanoylcarnitine or hydroxynonenoylcarnitine isomers	1.90	Acylcarnitine pathway
M49	C-Glycosyltryptophan	1.81	Tryptophan metabolism
M64	Methylbutyrylglycine or isovalerylglycine or valerylglycine	1.72	Glycine compound
M66	N2,N5-Diacetylornithine	1.71	Urea cycle, arginine and proline metabolism
M50	Indoleacetyl glutamine	1.65	Tryptophan metabolism
M67	1,7-Dimethyluric acid	1.64	Caffeine metabolism
M71	N-acetylaspartylglutamic acid	1.54	Alanine, aspartate and glutamate metabolism
M68	1,3,7-Trimethyluric acid	1.53	Caffeine metabolism
M72	Hexanoylglutamine	1.51	Glutamine metabolism
M44	Decanoylcarnitine or methylnonanoylcarnitine	1.49	Acylcarnitine pathway
M45	Nonenedioylcarnitine isomer	1.44	Acylcarnitine pathway
M31	γ-Glutamyl(iso)leucine	1.44	Leucine, isoleucine and valine metabolism
M73	Tyrosine	1.33	Tyr metabolism
M69	Caffeine	1.27	Caffeine metabolism
M56	Creatine	1.26	Animal protein consumption
M32	N-lactoyl(iso)leucine	1.22	Leucine, isoleucine and valine metabolism
M51	Kynurenic acid	1.18	Tryptophan metabolism
M52	5-Hydroxyindoleacetic acid	1.12	Tryptophan metabolism
M70	Paraxanthine	1.07	Caffeine metabolism

## Data Availability

The data presented in this study are available on request from the corresponding author. The data are not publicly available due to privacy restrictions.

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
