# Peer review of "Untargeted Metabolomic Study for Urinary Characterization of Adult Patients with Phenylketonuria"

_ijms, 2025, doi:10.3390/ijms262411808_

Round 1
Reviewer 1 Report
Comments and Suggestions for Authors
The application of untargeted urinary metabolomics in adults with phenylketonuria (PKU) is, in principle, a valuable approach. I appreciate the technical rigor of the metabolomic profiling and the authors’ interest in broadening our understanding of the PKU metabolic phenotype beyond phenylalanine. However, after careful review, I do not believe the current version of the manuscript meets the standards for publication.
Major Issues
-
Lack of Translational Applicability
Despite the use of a sophisticated analytical platform, the study does not lead to new or actionable biological or clinical insights. The primary finding—elevated phenylalanine and related metabolites—is expected and consistent with existing literature. Beyond this, the interpretation of the data is limited and remains largely descriptive. -
Weak Clinical Integration
The study lacks any integration with clinical, neurocognitive, behavioral, or nutritional outcomes. Several important aspects are mentioned—such as dietary implications, neurotoxicity, and disease severity—but none are explored using actual data. This significantly weakens the relevance of the metabolomic findings, as no associations are tested or validated. -
Disorganized and Speculative Discussion
The Discussion and Conclusions sections touch on many potentially interesting dimensions (nutrient imbalances, genotype–phenotype relationships, cognitive effects, severity classification), but these points are presented without structure or supporting data. As a result, the discussion feels speculative and disjointed rather than grounded in the study's findings.
Conclusion
While the methodological approach is strong, the manuscript lacks the clinical depth, analytical focus, and interpretive clarity needed to justify publication. I therefore recommend rejection in its current form. I encourage the authors to consider revising the study design in the future to incorporate clinical correlates that would allow the rich metabolomic data to be interpreted in a more meaningful and translationally relevant context.
Author Response
Comments 1: Lack of Translational Applicability -> Despite the use of a sophisticated analytical platform, the study does not lead to new or actionable biological or clinical insights. The primary finding—elevated phenylalanine and related metabolites—is expected and consistent with existing literature. Beyond this, the interpretation of the data is limited and remains largely descriptive.
Response 1: We appreciate the reviewer’s observation regarding the possible insights this study could offer. We believe that our findings provide meaningful biological insight, as the results are not limited to Phe and Phe-related metabolites. Specifically, 23% of the identified metabolites correspond to Phe metabolism, whereas the remaining 77% are related to other metabolic pathways, including tryptophan and leucine metabolism, purine metabolism, acylcarnitines, dietary and microbiota compounds. Following the reviewer’s comment, we have substantially revised the Results and Discussion Sections to highlight the relevance of the 77% of metabolites belonging to metabolic pathways other than Phe metabolism. In addition, the revised version has improved the contextualization and biological interpretation of the results reported. Moreover, we have placed particular emphasis on three compounds (8-Hydroxy-7-Methylguanine, N-acetyl(iso)leucine and 1H-Indole-3-carboxaldehyde) which are relevant in the multivariate ROC model. These findings expand current understanding of PKU by revealing alterations in multiple metabolic pathways.
All corresponding changes are highlighted in the revised manuscript.
Comments 2: Weak Clinical Integration -> The study lacks any integration with clinical, neurocognitive, behavioral, or nutritional outcomes. Several important aspects are mentioned—such as dietary implications, neurotoxicity, and disease severity—but none are explored using actual data. This significantly weakens the relevance of the metabolomic findings, as no associations are tested or validated.
Response 2: We appreciate the reviewer’s concern regarding the limited integration with several outcomes. Following the reviewer’s suggestion, we have now included additional information in table 1 (page 3) regarding their metabolic control (14 out of 36 patients with good metabolic control). We have also added nutritional parameters, including protein substitute intake for the PKU group and macronutrient distribution for both PKU and healthy controls (page 3). We believe that these additions significantly improve the clinical relevance of the metabolomic findings reported.
Furthermore, we consider our study relevant for providing a robust metabolomic characterization of adults with PKU and for demonstrating the involvement of multiple metabolic pathways, as well as the dietary and biological insights derived from this PKU population.
As examples of new relevant biological insights, we have added:
- …”8-Hydroxy-7-methylguanine (M18) was identified as one of the most discriminant metabolites in our cohort, showing the highest AUC value (0.99) in the univariate ROC analysis (Figure 3A) and being consistently selected in the Random Forest models (Figure 4). This compound has been previously detected in human urine by Liu et al. in their study among healthy adults and children. Biochemically, 8-Hydroxy-7-methylguanine originates from the oxidation of 7-MG, a reaction catalyzed by the enzyme xanthine oxidase. Since 7-MG has been described as a marker of DNA damage, as it can induce DNA strand breaks, its oxidation may reflect altered nucleotide metabolism or oxidative stress. In this sense, Dobrowolski et al. observed aberrant DNA methylation in patients with PKU, with a more extensive methylation in patients with high Phe exposure. To our knowledge, this is the first report of elevated urinary 8-Hydroxy-7-methylguanine (M18) in PKU, suggesting that purine metabolism and oxi-dative processes may contribute to the metabolic complexity of PKU and indicates the need for further research to assess its potential as a biomarker. (Page 12, lines 370-383)
The tryptophan metabolism has also been rewritten:
- An important biological insight of our cohort is the alteration of Trp metabolism, with both up- and downregulated metabolites (14% of identified metabolites) across its main pathways. Trp catabolism can be divided in: (a) the indole and tryptamine pathway, (b) serotonin pathway, and (c) kynurenine pathway. (a) indole metabolite (M21-M23) concentrations, a microbiota-derived metabolite, were higher in patients with PKU. Notably, 1H-Indole-3-carboxaldehyde (M21) showed high potential as a marker in both univariate (Figure 3) and multivariate ROC curves analyses (Figure 4D). This reinforces growing evidence in gut-brain axis interactions in PKU, where microbiota-derived metabolites might play a role in clinical manifestations. (Page 12, lines 384-392).
Implications of dietary metabolites with microbiota:
- “…The lower levels of microbiota-derived phenolic compounds (M58–M62) could indicate altered gut microbiome activity in PKU, consistent with studies describing com-positional and functional differences in the gut microbiota of this population”. (Page 14, lines 457-460).
All the corresponding changes are highlighted in the revised manuscript.
Comments 3: Disorganized and Speculative Discussion -> The Discussion and Conclusions sections touch on many potentially interesting dimensions (nutrient imbalances, genotype–phenotype relationships, cognitive effects, severity classification), but these points are presented without structure or supporting data. As a result, the discussion feels speculative and disjointed rather than grounded in the study's findings.
Response 3: We agree with the reviewer that the discussion would benefit with a better structure, and should also be grounded in the study’s findings. Following the reviewer’s comment, we have refined the Discussion section of the manuscript and have also improved the Conclusions section.
Specifically, we have refined its content by removing some paragraphs and sentences that were less directly relevant to our study results. Specifically, we have condensed the section discussing Phe-related metabolites, removed the paragraph on pterin compounds for not being so relevant for our study results (now integrated into the ROC analysis section), and deleted the Glycine and enrichment analysis paragraphs, which were partly redundant with the reported findings. In addition, we have reorganized several sections and added new discussion about some key metabolites (8-Hydroxy-7-methylguanine, metabolites related to tryptophan metabolism, and diet- or microbiota-derived compounds) to improve readability and modified certain parts to strengthen the alignment between the discussion and our results.
The changes are highlighted in the revised version of the manuscript.
Response to Comments on the Quality of English Language
Point 1: The English is fine and does not require any improvement.
Response 1: We thank the reviewer for the positive evaluation of the English quality.
Additional clarifications
Additionally, we have corrected the reported values for Total protein intake, Vitamin E intake and Phe in Table 1, as the previous version displayed mean instead of median values.

Reviewer 2 Report
Comments and Suggestions for Authors
Gonzalez-Rodriguez et al. report metabolic features which are statistically significant between the urine of PKU patients following a defined diet and healthy controls, using a LC-MS based metabolomics approach. The methods are not new but according to authors the novelty relies on the metabolic features found discriminating these two groups. It also seems that urinary metabolomics hasn’t been that much studied for this particular metabolic disorder. While the study itself is interesting and the description of the (potential) metabolites is extensive and well thought of, I cannot recommend the publication of this piece in IJMS in its current form. I have two major concerns:
- The experimental design might be flawed: as authors indicate on line 111 and on Table 1, they match PKU patients following a controlled diet and healthy controls based on different demographics. However, since the diet of the control group was not controlled, this is a big confounding factor. I believe authors cannot correct this and as such the results are dubious.
- Authors refer to their statistically significant features as biomarkers however clinical validation has not been conducted (e.g. replication in independent cohort). Additionally, I wonder what the real value is. Do we really need more “urinary markers“ for this disorder? Considering that PKU is well characterized by elevated Phe levels and this can be done relatively straightforward during screening methods. This has to be better presented.
Other key comments for discussion:
- Title and abstract: authors should refrain from using the word novel. It is implicitly understood that the findings of a research article are novel thus it is redundant.
- Figure 1 shows the PCA of the LC-MS ESI+ and ESI- metabolomics. Why are the water blanks and the standards included in the PCA? This severely bias the results (e.g. now the PC2 only covers 3-4 % of the variability since the QC1/QC2 are driving the group separation). This is not common practice. I also wonder how many metabolic features were found in QC1 and QC2?
- Fig S1: there is no need to include a 2D snapshot of a 3D plot. This can be misleading because it obscures the true relationship between the presented observations. Including a 3D plot would only make sense if authors make it interactive so that readers can explore the data themselves without being offered a preferred angle of projection.
- Authors should better explain how to interpret the plots from Figure S3 and S4.
Other (minor) comments:
- Lines 41 and 102: adults treated with what exactly?
- What was the re-equilibration time of the column?
- In the MS/MS fragmentation column (Table S1), for M01, M22 and M23 authors included the (de)protonated monoisotopic mass, however, these are technically not fragments. Additionally, how would authors assign a MSI ID level of 1 if the MS/MS fragments are not generated? Is this perhaps a mistake? Similar findings are observed on Table S2, e.g. M47, M48 and M51 among others. Please, check.
- Tables 2 and 3: authors mention these metabolites are ordered by VIP values. It is strange that the actual VIP values are not included also in this table.
- Discussion section is extremely long. Authors should consider condensing it.
Author Response
Comments 1: The experimental design might be flawed: as authors indicate on line 111 and on Table 1, they match PKU patients following a controlled diet and healthy controls based on different demographics. However, since the diet of the control group was not controlled, this is a big confounding factor. I believe authors cannot correct this and as such the results are dubious.
Response 1: Thank you for pointing this out. We appreciate the reviewer’s concern about the experimental design. However, we believe it may have been a misunderstanding. We have revised the paragraph of results (Section 2.1 - Clinical characteristics of the study population, now in lines 109-113) and we have specified that no statistically significant differences were observed in demographics (sex, age, BMI) between control participants and PKU subjects. We have removed that information that may have caused a misinterpretation of the data.
This study follows an observational case-control design in accordance with the STROBE guidelines [1]. Therefore, controls were included to establish a valid reference group, as done in previous metabolomic studies in PKU populations that did not strictly control the diet of healthy participants [2,3,4].
To clarify this statement, we revised the first paragraph of Section 4.1 (Subjects and study design) at page 14, line 487-488, to explicitly describe how controls were selected: “To ensure comparability, participants were matched to individuals with PKU according to age, sex, and BMI”.
We thank the reviewer for the insightful comment regarding the dietary characteristics of the control group. This is an important aspect that was not correctly addressed in the original version of the manuscript. Following the reviewer’s comment, we have added macronutrient intake data for both groups to Table 1. We have observed that the macronutrient distribution was consistent with that reported for the general Spanish adult population [5,6]. Moreover, the dietary intake data of both PKU and control participants, are in line with previous studies that included comparable control populations [7,8].
This clarification is included in the revised version:
- 1. Clinical characteristics of the study population: “…In addition, differences in macronutrient composition were observed as fat intake was higher in the control group, whereas carbohydrate intake was significantly higher in the PKU group. These differences likely reflect distinct dietary habits influenced by the protein restriction inherent to PKU management as previously already observed in other studies comparing dietary habits of PKU and matched controls.” (page 3, lines 116-121).
- Discussion Section: “…This pattern aligns with the dietary data, showing lower meat consumption and reduced natural protein intake among individuals with PKU (Table 1), despite similar total protein and energy intake between groups. It is important to note that the diet of the control population was consistent with the macronutrient distribution reported for the general Spanish adult population”. (Page 14, lines 450-454):
- von Elm E, Altman DG, Egger M, Pocock SJ, Gotzsche PC, Vandenbroucke JP. The Strengthening the Reporting of Observational Studies in Epidemiology (STROBE) Statement: guidelines for reporting observational studies. Available at: https://www.equator-network.org/reporting-guidelines/strobe/
- Cannet, C.; Bayat, A.; Frauendienst-Egger, G.; Freisinger, P.; Spraul, M.; Himmelreich, N.; Kockaya, M.; Ahring, K.; Godejohann, M.; MacDonald, A.; et al. Phenylketonuria (PKU) Urinary Metabolomic Phenotype Is Defined by Genotype and Metabolite Imbalance: Results in 51 Early Treated Patients Using Ex Vivo 1H-NMR Analysis. Molecules 2023, 28, 4916, doi:10.3390/MOLECULES28134916/S1.
- Mütze, U.; Beblo, S.; Kortz, L.; Matthies, C.; Koletzko, B.; Bruegel, M.; Rohde, C.; Thiery, J.; Kiess, W.; Ceglarek, U. Metabolomics of Dietary Fatty Acid Restriction in Patients with Phenylketonuria. PLoS One 2012, 7, e43021, doi:10.1371/journal.pone.0043021.
- Schoen, M.S.; Singh, R.H. Plasma Metabolomic Profile Changes in Females with Phenylketonuria Following a Camp Intervention. Am J Clin Nutr 2022, 115, 811–821, doi:10.1093/ajcn/nqab400.
- del Pozo de la Calle, S.; García Iglesias, V.; Cuadrado Vives, C.; Ruiz Moreno, E.; Valero Gaspar, T.; Ávila Torres, J.M.; Varela Moreiras, G. Valoración Nutricional de La Dieta Española de Acuerdo al Panel de Consumo Alimentario; Fundación Española de La Nutrición (FEN), Spain, 2012; pp. 1-140.
- Ruiz, E.; Ávila, J.M.; Valero, T.; del Pozo, S.; Rodriguez, P.; Aranceta-Bartrina, J.; Gil, Á.; González-Gross, M.; Ortega, R.M.; Serra-Majem, L.; et al. Macronutrient Distribution and Dietary Sources in the Spanish Population: Findings from the ANIBES Study. Nutrients 2016, 8, doi:10.3390/NU8030177.
- Evans, S.; Daly, A.; Wildgoose, J.; Cochrane, B.; Chahal, S.; Ashmore, C.; Loveridge, N.; Macdonald, A. Growth, Protein and Energy Intake in Children with PKU Taking a Weaning Protein Substitute in the First Two Years of Life: A Case-Control Study. Nutrients 2019, 11, doi:10.3390/nu11030552.
- Sailer, M.; Elizondo, G.; Martin, J.; Harding, C.O.; Gillingham, M.B. Nutrient Intake, Body Composition, and Blood Phenylalanine Control in Children with Phenylketonuria Compared to Healthy Controls. Mol Genet Metab Rep 2020, 23, doi:10.1016/j.ymgmr.2020.100599.
Comments 2: Authors refer to their statistically significant features as biomarkers however clinical validation has not been conducted (e.g. replication in independent cohort). Additionally, I wonder what the real value is. Do we really need more “urinary markers” for this disorder? Considering that PKU is well characterized by elevated Phe levels and this can be done relatively straightforward during screening methods. This has to be better presented.
Response 2: Thank you for this comment. We performed ROC curve analyses to identify potential biomarkers for classifying our study population, following metabolomics recommendations [1] and our previous experience in biomarker assessment [2,3], as this approach is considered one of the most objective and statistically valid method for evaluating biomarker performance. We focused on those metabolites that had AUCs values higher than 0.9, which its utility as biomarker is classified as excellent [1]. Moreover, these selected metabolites may help identify alterations in metabolic pathways and contribute to understanding PKU pathophysiology, and may help reveal new therapeutic targets [4]. As the reviewer suggests, the metabolites should be validated in an independent cohort. We agree with the reviewer that referring to a metabolite directly as a biomarker may seem too ambitious, and therefore, we have replaced them with “potential biomarker” or with compound or metabolite where appropriate.
The reviewer is correct that Phe and its related metabolites are already well-established biomarkers used in PKU screening. In our cohort, however, only 23% of the identified metabolites were associated with Phe metabolism, while the remaining 77% belonged to other pathways, including leucine and tryptophan metabolism, acylcarnitines, vitamins, and diet- or microbiota-derived compounds. Following the reviewer’s suggestion, we have clarified this aspect throughout the revised manuscript. Furthermore, in the Random Forest models, three metabolites from other pathways—specifically related to purine, leucine, and tryptophan metabolism—showed strong discriminative power, further illustrating the broader metabolic disturbances characteristic of PKU.
While blood Phe levels remains key for PKU screening and diagnosis, exploring additional metabolites can provide complementary insights. Such metabolites could be valuable in future studies assessing clinical, neurocognitive and behavioral outcomes, since blood Phe concentration does not fully capture the metabolic complexity of PKU and may not always be sufficiently reliable as the only outcome predictor in PKU [5]. Few studies have analyzed more than ten urinary metabolites in PKU compared with healthy controls [6,7,8], and we believe this is an area that would greatly benefit from our presented research and would add valuable complementary information to better characterize the urinary metabolomic fingerprint of adults with PKU.
Therefore, we have implemented several modifications in the revised manuscript (see the version with changes highlighted in yellow). Some of the main revisions include the following:
- Discussion section: “…revealing a wide range of metabolic alterations. Although Phe and Phe-related com-pounds accounted for 23% of the identified metabolites, the majority of identified metabo-lites (77%) belonged to other pathways, including those related to several amino acids, acylcarnitines, micronutrients, and dietary and microbial metabolites. These findings highlight the complex metabolic impact of PKU and provide biological insights that may influence disease progression and clinical outcomes.” (Page 11-12, 340-345).
- 3. Identification of differential metabolites: “A detailed list of the identification parameters, including mass fragmentation, time retention and statistical values, is provided for each metabolite in Table S1 and Table S2. In total, 29 metabolites were upregulated and 44 were downregulated in patients with PKU. Table 2 and Table 3 present these metabolites in decreasing order of VIP value for the upregulated and downregulated compounds, respectively. When categorized by metabol-ic pathway, 18 compounds were associated to dietary, microbiota, and micronutrient in-take; 17 belonged to Phe metabolism; 14 to carnitine metabolism; 10 to tryptophan (Trp) metabolism; 3 to leucine (Leu) metabolism; and 2 to pteridine metabolism, among others.” (Page 5, lines 157-164).
- We have changed the title of the 2.5 section. We have changed Diagnostic models for ROC analysis. (Page 9, line 259).
- 5. ROC analysis: “…Among these, 8-Hydroxy-7-methylguanine (M18) showed the highest AUC value, followed by Phe (M01) and other Phe-related metabolites. Additional compounds such as N-acetyl(iso)leucine (M30), isoxanthopterin (M19), 1H-Indole-3-carboxaldehyde (M21), and heptenoylcarnitine (M33) also demonstrated high AUC values.” (Page 9, lines 264-267)
- Xia, J.; Broadhurst, D.I.; Wilson, M.; Wishart, D.S. Translational Biomarker Discovery in Clinical Metabolomics: An Introductory Tutorial. Metabolomics 2013, 9, 280–299, doi:10.1007/s11306-012-0482-9.
- Urpi-Sarda, M.; Almanza-Aguilera, E.; Llorach, R.; Vázquez-Fresno, R.; Estruch, R.; Corella, D.; Sorli, J. V.; Carmona, F.; Sanchez-Pla, A.; Salas-Salvadó, J.; et al. Non-Targeted Metabolomic Biomarkers and Metabotypes of Type 2 Diabetes: A Cross-Sectional Study of PREDIMED Trial Participants. Diabetes Metab 2019, 45, 167–174, doi:10.1016/j.diabet.2018.02.006.
- Garcia-Aloy, M.; Llorach, R.; Urpi-Sarda, M.; Tulipani, S.; Estruch, R.; Martínez-González, M.A.; Corella, D.; Fitó, M.; Ros, E.; Salas-Salvadó, J.; et al. Novel Multimetabolite Prediction of Walnut Consumption by a Urinary Biomarker Model in a Free-Living Population: The Predimed Study. J Proteome Res 2014, 13, 3476–3483, doi:10.1021/pr500425r.
- Qiu, S.; Cai, Y.; Yao, H.; Lin, C.; Xie, Y.; Tang, S.; Zhang, A. Small Molecule Metabolites: Discovery of Biomarkers and Therapeutic Targets. Signal Transduct Target Ther 2023, 8.
- van Wegberg, A.M.J.; van der Weerd, J.C.; Engelke, U.F.H.; Coene, K.L.M.; Jahja, R.; Bakker, S.J.L.; Huijbregts, S.C.J.; Wevers, R.A.; Heiner-Fokkema, M.R.; van Spronsen, F.J. The Clinical Relevance of Novel Biomarkers as Outcome Parameter in Adults with Phenylketonuria. J Inherit Metab Dis 2024, doi:10.1002/jimd.12732.
- Blasco, H.; Veyrat-Durebex, C.; Bertrand, M.; Patin, F.; Labarthe, F.; Henique, H.; Emond, P.; Andres, C.R.; Antar, C.; Landon, C.; et al. A Multiplatform Metabolomics Approach to Characterize Plasma Levels of Phenylalanine and Tyrosine in Phenylketonuria. JIMD Rep 2017, 32, 69–79, doi:10.1007/8904_2016_568.
- Xiong, X.; Sheng, X.; Liu, D.; Zeng, T.; Peng, Y.; Wang, Y. A GC/MS-Based Metabolomic Approach for Reliable Diagnosis of Phenylketonuria. Anal Bioanal Chem 2015, 407, 8825–8833, doi:10.1007/s00216-015-9041-3.
- Cannet, C.; Bayat, A.; Frauendienst-Egger, G.; Freisinger, P.; Spraul, M.; Himmelreich, N.; Kockaya, M.; Ahring, K.; Godejohann, M.; MacDonald, A.; et al. Phenylketonuria (PKU) Urinary Metabolomic Phenotype Is Defined by Genotype and Metabolite Imbalance: Results in 51 Early Treated Patients Using Ex Vivo 1H-NMR Analysis. Molecules 2023, 28, 4916, doi:10.3390/MOLECULES28134916/S1.
Comments 3: Title and abstract: authors should refrain from using the word novel. It is implicitly understood that the findings of a research article are novel thus it is redundant.
Response 3: We appreciate the reviewer’s observation regarding the redundancy of the word novel. Accordingly, we have removed the word novel from the abstract and from other parts of the manuscript. Moreover, we believe that due to the changes made to the manuscript, the article would benefit from a revised title. Therefore, the revised title is now “Untargeted Metabolomic Study for Urinary Characterization of Adult Patients with Phenylketonuria”.
Comments 4: Figure 1 shows the PCA of the LC-MS ESI+ and ESI- metabolomics. Why are the water blanks and the standards included in the PCA? This severely bias the results (e.g. now the PC2 only covers 3-4 % of the variability since the QC1/QC2 are driving the group separation). This is not common practice. I also wonder how many metabolic features were found in QC1 and QC2?
Response 4: The reviewer is right to point out it. We previously included QC1 and QC2 in the PCA as part of the analytical quality assessment, following previous methodology [1,2]. However, following the reviewer’s recommendation and to avoid potential misinterpretation, we have removed this Figure from the manuscript.
- González-Domínguez, Á.; Estanyol-Torres, N.; Brunius, C.; Landberg, R.; González-Domínguez, R. QComics: Recommendations and Guidelines for Robust, Easily Implementable and Reportable Quality Control of Metabolomics Data. Anal Chem 2024, 96, 1064–1072, doi:10.1021/acs.analchem.3c03660.
- Llorach, R.; Urpi-Sarda, M.; Jauregui, O.; Monagas, M.; Andres-Lacueva, C. An LC-MS-Based Metabolomics Approach for Exploring Urinary Metabolome Modifications after Cocoa Consumption. J Proteome Res 2009, 8, 5060–5068, doi:10.1021/pr900470a.
Comments 5: Fig S1: there is no need to include a 2D snapshot of a 3D plot. This can be misleading because it obscures the true relationship between the presented observations. Including a 3D plot would only make sense if authors make it interactive so that readers can explore the data themselves without being offered a preferred angle of projection.
Response 5: We agree with the reviewer that the 3D plot may be misleading. As suggested by the reviewer, we have removed Figure S1C and S1D at page 20, line 95 of the Supplementary materials.
Comments 6: Authors should better explain how to interpret the plots from Figure S3 and S4.
Response 6: We appreciate the reviewer’s comment. Permutation tests already provide a robust validation of the OPLS-DA model, and the model diagnostics (Figure S3 and S4) would only serve as complementary information to figure S2. R2Y or R2X represents the explanation of variance, while Q2 represents the predictability of the model, and the closer to 1 the better the model is [1]. Therefore, to improve the clarity of the article, we removed these 2 figures and kept the permutation test results (Figure S2 of the revised supplementary material).
- Zhang, R.Z.; Zhao, J.T.; Wang, W.Q.; Fan, R.H.; Rong, R.; Yu, Z.G.; Zhao, Y.L. Metabolomics-Based Comparative Analysis of the Effects of Host and Environment on Viscum Coloratum Metabolites and Antioxidative Activities. J Pharm Anal 2022, 12, 243–252, doi:10.1016/j.jpha.2021.04.003.
Comments 7: Lines 41 and 102: adults treated with what exactly?
Response 7: We agree with the reviewer that the treatment followed by patients with PKU was not clearly described in the original version. The dietary treatment approach for PKU patients in our cohort is consistent with the European PKU guidelines [1].
To ensure clarity, we have included the treatment of PKU individuals in the “Materials and Methods” section (page 14, lines 482-484) “Patients with PKU were under nutritional counseling and clinical monitoring, following dietary recommendations that included the use of PS and a low-Phe diet. None of the patients were receiving BH4 treatment”.
- van Wegberg, A.M.J.; MacDonald, A.; Ahring, K.; Bélanger-Quintana, A.; Beblo, S.; Blau, N.; Bosch, A.M.; Burlina, A.; Campistol, J.; CoÅŸkun, T.; et al. European Guidelines on Diagnosis and Treatment of Phenylketonuria: First Revision. Mol Genet Metab 2025, 145.
Comments 8: What was the re-equilibration time of the column?
Response 8: Thank you for this comment. Following the reviewer’s suggestion, we have added the sentence “The re-equilibration time of the column was 4 min.” on page 15, in the paragraph of Section 4.3 (HPLC-QTOF-MS) at line 522.
Comments 9: In the MS/MS fragmentation column (Table S1), for M01, M22 and M23 authors included the (de)protonated monoisotopic mass, however, these are technically not fragments. Additionally, how would authors assign a MSI ID level of 1 if the MS/MS fragments are not generated? Is this perhaps a mistake? Similar findings are observed on Table S2, e.g. M47, M48 and M51 among others. Please, check.
Response 9: Thank you for pointing this out. We had previously only included the parent ion for those metabolites with available standards (MSI ID level 1). As suggested by the reviewer, we have updated the MS/MS fragments for the corresponding metabolites in Table S1 (M01, M22, M23) and Table S2 (M47, M48, M51, M53, M55, M56, M69 and M73).
Comments 10: Tables 2 and 3: authors mention these metabolites are ordered by VIP values. It is strange that the actual VIP values are not included also in this table.
Response 10: We completely agree with this comment. Accordingly, we have added an additional column in table 2 and 3 displaying the corresponding VIP values for each metabolite. Moreover, we have reorganized both tables to order the metabolites by decreasing VIP value, rather than grouping them by biological pathway. This change better supports our results, emphasizing that pathway alterations such as purine metabolism (DNA), tryptophan metabolism, acylcarnitines, leucine metabolism, and dietary compounds, among others, play a relevant role in our high-quality OPLS-DA models and have a strong impact in the metabolomic fingerprint of PKU.
Comments 11: Discussion section is extremely long. Authors should consider condensing it.
Response 11: Following the recommendations of the reviewer we have revised this part. Although the revised version maintains a similar overall length, we have refined its content by removing paragraphs and sentences that were less directly relevant to our study results. Specifically, we have condensed the section discussing Phe-related metabolites, removed the paragraph on pterin compounds (now integrated into the ROC analysis section), and deleted the Glycine and enrichment analysis paragraphs, which were partly redundant with the reported findings. In addition, we have reorganized several sections to improve readability and modified certain parts to strengthen the alignment between the discussion and our results.
These changes are highlighted in the revised manuscript.
Response to Comments on the Quality of English Language
Point 1: The English is fine and does not require any improvement.
Response 1: We thank the reviewer for the positive evaluation of the English quality.
Additional clarifications
Additionally, we have corrected the reported values for Total protein intake, Vitamin E intake and Phe in Table 1, as the previous version displayed mean instead of median values.

Round 2
Reviewer 1 Report
Comments and Suggestions for Authors
The authors presented a much better version of the manuscript, having addressed most of the issues I have raised in the first review.
However, one major issue still remains in the Discussion. The authors inadvertently mixed together their results based on the metabolomics of urine samples with other authors who did plasma or dried blood spot metabolomics studies.
When discussing their findings on tryptophan metabolites from the kynurenine pathway, the authors compare them with those of Schoen and Singh (2022) and Boulet et al (2020). However, both articles dealt with metabolomics or LC/MS-MS of plasma samples. The authors did not unfortunately warn the readers. This is not an irrelevant question, as kynurenine pathway metabolites can have disparately different levels in blood and urine.
In the discussion of reduced levels of isoleucine metabolites, the authors refer the articles of Schoen and Singh (2022) and Blasco et al (2017). Although the latter included plasma and urine samples, the results of isoleucine were restricted to plasma. Again the authors did not alert the readers about the samples used in these references; moreover, the role of isoleucine and other large neutral amino acids (LNAA) in competing with phenylalanine for brain uptake must be investigated in blood or even cerebrospinal fluid - even though the latter is obtained by invasive procedures. However, the authors should alert that urine levels of isoleucine metabolites may reflect nitrogen balance rather than the LNAA concentrations at the level of the blood/brain barrier. For any metabolites that are dependent on blood/brain barrier gradients for its physiological or pathogenic effect, their consequences cannot be inferred by urine concentrations.
Regarding acylcarnitines, the papers referenced by the authors - Schoen and Singh (2022), Mütze et al (2012), and Weigel et al (2008), used plasma or dried blood spots. I am not saying that urinary acylcarnitines do not have a potential role in PKU, but the authors in the paragraph 426-436 did not make clear what this might be.
Minor: the nomenclature adopted for the compounds are sometimes not standard, for example N-acetyl(iso)leucine (415) and (iso)leucine (416) instead of N-acetylisoleucine and isoleucine; improper use of capital letters for the name of compounds not in the beginning of a sentence - L-Carnitine (Table 1, 211), 2-Hydroxyphenylacetic acid (348), Phenylacetylglutamine (349).
Minor: paragraph 384-413, bullets (a), (b), (c) are repeated as the authors are describing further compounds in each category, but this may not be clear to the reader.
Author Response
Comments 1: The authors presented a much better version of the manuscript, having addressed most of the issues I have raised in the first review.
However, one major issue still remains in the Discussion. The authors inadvertently mixed together their results based on the metabolomics of urine samples with other authors who did plasma or dried blood spot metabolomics studies.
Response 1: Thank you for this important feedback. We have now revised the Discussion to specify whether previously published findings were obtained from urine, plasma, or dried blood spot to avoid unintended mixing of biospecimen types. This clarification ensures that comparisons are appropriately contextualized.
In addition, to strengthen the interpretation of our urinary findings, we included complementary analyses of plasma samples from the same individuals for metabolites in pathways where additional interpretation was deemed beneficial (tryptophan metabolism, leucine metabolism, and acylcarnitines). This has allowed us to assess correlations between urine and plasma metabolites and to establish the biological relevance of our results. Overall, plasma results were consistent with those observed in urine. Not all metabolites were detected in plasma, as our methodology has an untargeted approach and therefore neither as specific nor as sensitive as a metabolomic targeted analysis, which was beyond the scope of this study.
Accordingly, the manuscript highlights the following changes (in yellow): the plasma sample preparation protocol (Supplementary Information; lines 53-61), updates to the Materials and Methods section (page 15-16), and the statistical analysis parameters (page 15-16) for group comparisons and correlation analyses between plasma and urine results.
Comments 2: When discussing their findings on tryptophan metabolites from the kynurenine pathway, the authors compare them with those of Schoen and Singh (2022) and Boulet et al (2020). However, both articles dealt with metabolomics or LC/MS-MS of plasma samples. The authors did not unfortunately warn the readers. This is not an irrelevant question, as kynurenine pathway metabolites can have disparately different levels in blood and urine.
Response 2: Thank you for this comment. We have revised the Discussion to indicate the type of samples analyzed in the referenced studies.
Discussion (page 13, lines 396-398):
- “…Schoen et al. [22] also reported plasma reduced levels of kynurenine (M48) and Trp (M47) in females with PKU and poor metabolic control. Boulet et al. [40] similarly observed a disruption in plasma Trp metabolism in patients with PKU, with downregulation of kynurenine (M48), upregulation of kynurenic acid (M51), and no significant differences in Trp (M47)…”
In addition, to strengthen the interpretation of our urinary findings, we included complementary plasma data for this pathway. These results are now integrated into the manuscript and help reinforce the biological coherence of the Trp-related alterations observed in urine.
Results (2.3.4. Tryptophan and tryptophan metabolism compounds, page 6, lines 202-212):
- “…Given the growing interest in these pathways in PKU, we further examined the relation-ship between urine and plasma levels for metabolites detected in both biological samples. Significant positive correlations were observed for 1H-indole-3-carboxaldehyde (r = 0.751, p <0.001), indolelactic acid (r = 0.759, p <0.001), N,N,N-trimethyltryptophan betaine (r = 0.834, p <0.001), kynurenine (r = 0.414, p <0.001), and Trp (r = 0.300, p = 0.0497), with consistent PKU–control differences in urine and plasma (p <0.05). As expected from their shared metabolic origin, urinary kynurenine (M48) and urinary kynurenic acid (M51) were strongly correlated (r = 0.639, p < 0.001), and urinary kynurenic acid was also correlated with plasma kynurenine (r = 0.425, p < 0.001), supporting coordinated alterations in the kynurenine pathway across different matrices.”
These positive urinary-plasma correlations and consistent group differences strengthen the robustness of our findings and support the biological relevance of the urinary alterations reported.
The changes are highlighted in the revised version of the manuscript.
Comments 3: In the discussion of reduced levels of isoleucine metabolites, the authors refer the articles of Schoen and Singh (2022) and Blasco et al (2017). Although the latter included plasma and urine samples, the results of isoleucine were restricted to plasma. Again, the authors did not alert the readers about the samples used in these references; moreover, the role of isoleucine and other large neutral amino acids (LNAA) in competing with phenylalanine for brain uptake must be investigated in blood or even cerebrospinal fluid - even though the latter is obtained by invasive procedures. However, the authors should alert that urine levels of isoleucine metabolites may reflect nitrogen balance rather than the LNAA concentrations at the level of the blood/brain barrier. For any metabolites that are dependent on blood/brain barrier gradients for its physiological or pathogenic effect, their consequences cannot be inferred by urine concentrations.
Response 3: Thank you for this comment. We have revised the Discussion to clearly specify the type of samples analyzed in the referenced studies.
Discussion (page 13, lines 409-410):
- “…Both Leu and Ile have been previously found in metabolomic studies, analyzing plasma samples, to be downregulated in PKU [22,42]…”
In line with the reviewer’s suggestion, we have clarified in the Discussion that urinary LNAA-related metabolites cannot be directly interpreted in relation to blood–brain barrier transport. To address this point, we added a sentence emphasizing the need for further investigation into their systemic relevance:
Discussion (page 13, line 410, 415-417):
- “…Elevated blood LNAA concentrations may reduce Phe transport into the brain [41]…”
- “…Additional research is needed to clarify how alterations in urinary leucine-derived metabolites relate to systemic LNAA metabolism and nutritional status.”
In addition, to strengthen the interpretation of our urinary findings, we incorporated complementary plasma data for this pathway. We found that plasma leucine concentrations (quantified during routine aminogram analysis) were significantly lower in patients with PKU. Importantly, plasma leucine showed a positive and statistically significant correlation with urinary N-acetyl(iso)leucine (M30), suggesting that the urinary signal of M30 may reflect an alteration in leucine metabolism rather than nitrogen balance alone.
Results (2.1. Clinical characteristics of the study population, page 4, Table 1)
- Leucine and Isoleucine median levels with corresponding p-value.
Results (2.3.5. Leucine-derived compounds, page 6, lines 216-221):
- “…Consistent with these urinary findings, plasma leucine concentrations were significantly lower in patients with PKU compared with controls (p <0.001), while plasma isoleucine only showed a trend (p = 0.062). In addition, we observed a positive correlation between plasma leucine levels and urinary N-acetyl(iso)leucine (M30) (r = 0.34, p = 0.009), supporting that this compound could be a leucine metabolite.”
The changes are highlighted in the revised version of the manuscript.
Comments 4: Regarding acylcarnitines, the papers referenced by the authors - Schoen and Singh (2022), Mütze et al (2012), and Weigel et al (2008), used plasma or dried blood spots. I am not saying that urinary acylcarnitines do not have a potential role in PKU, but the authors in the paragraph 426-436 did not make clear what this might be.
Response 4: We thank the reviewer for this important observation; we have revised the discussion to specify the type of sample analyzed in acylcarnitine studies.
To address this feedback, we have made the following changes to its paragraph in the discussion section:
Discussion (page 13, lines 419-421 and 424-427):
- “Acylcarnitines represented another major group of altered metabolites in this PKU population (19% of total metabolites). We observed increased levels of both urinary and plasma phenylacetylcarnitine (M24) in patients with PKU, as well as decreased levels of 13 urinary acylcarnitines (M33-M45), with decreased plasma levels of M36. Although free carnitine deficiency is commonly reported in PKU and has been associated with increased oxidative stress [44–46], no such deficiency was observed in our cohort, which may have been influenced by supplementation or cohort characteristics. Previous studies describing acylcarnitine alterations in PKU have primarily analyzed plasma [22] or dried blood spots [47,48], reporting both upregulation (including phenylacetylcarnitine) [22] and downregulation of various acylcarnitines species [47,48]. Changes in acylcarnitines profiles have been linked to impaired energy metabolism [48] and have been proposed to play a role in neurotransmission [49]. Further investigation into the role of specific carnitine derivatives identified is warranted to clarify their contribution to PKU pathophysiology.”
In addition, we have added complementary plasma data to reinforce the urinary findings.
Results (2.3.6. Carnitine metabolites, page 6-7, lines 230-237):
- “…Importantly, urinary phenylacetylcarnitine (M24) showed a positive correlation with its plasma levels (r = 0.588, p <0.001), also being upregulated in patients with PKU. The other acylcarnitines identified in this study showed lower levels in patients with PKU. In particular, the acylcarnitine M36—putatively annotated as undecanoylcarnitine, dimethyl-nonanoylcarnitine, or methyldecanoylcarnitine—was reduced in both urine and plasma of patients with PKU, a pattern further supported by a positive correlation between the two matrices (r = 0.361, p = 0.002). This indicates that the decrease observed in urine is mirrored by the same direction of change in plasma.”
The changes are highlighted in the revised version of the manuscript.
Comments 5: Minor: the nomenclature adopted for the compounds are sometimes not standard, for example N-acetyl(iso)leucine (415) and (iso)leucine (416) instead of N-acetylisoleucine and isoleucine; improper use of capital letters for the name of compounds not in the beginning of a sentence - L-Carnitine (Table 1, 211), 2-Hydroxyphenylacetic acid (348), Phenylacetylglutamine (349).
Response 5: We agree with the reviewer that some compound names required standardization. Regarding the use of “N-acetyl(iso)leucine” and “(iso)leucine” examples, we adopted this nomenclature because our analytical method does not allow discrimination between the leucine and isoleucine isomers, and this nomenclature is consistent with previously published studies (1). Although the positive correlation between urinary N-acetyl(iso)leucine (M30) and plasma leucine supports the interpretation that this metabolite may more likely correspond to N-acetylleucine rather than N-acetylisoleucine, we have retained the annotation “N-acetyl(iso)leucine” in accordance with MSI level 2 identification standards.
Regarding the use of capital letters for the name of compounds not in the beginning of a sentence, we have corrected the use of capital letters for these metabolites across the manuscript.
- Jansen, R.S.; Addie, R.; Merkx, R.; Fish, A.; Mahakena, S.; Bleijerveld, O.B.; Altelaar, M.; IJlst, L.; Wanders, R.J.; Borst, P.; et al. N-Lactoyl-Amino Acids Are Ubiquitous Metabolites That Originate from CNDP2-Mediated Reverse Proteolysis of Lactate and Amino Acids. Proceedings of the National Academy of Sciences 2015, 112, 6601–6606, doi:10.1073/PNAS.1424638112.
Comments 6: Minor: paragraph 384-413, bullets (a), (b), (c) are repeated as the authors are describing further compounds in each category, but this may not be clear to the reader.
Response 6: We agree with the reviewer that the inclusion of bullets may be confusing to the reader and therefore, we have removed it from the revised manuscript.

Reviewer 2 Report
Comments and Suggestions for Authors
Authors have clearly improved their manuscript. I enjoyed reading this new updated version. I have no further comments.
Author Response
We would like to sincerely thank the reviewer for their positive evaluation of our revised manuscript. We greatly appreciate their insightful comments and constructive feedback, which have contributed to improving the quality of our work.